# Teachers’ and Principals’ Familiarity with School Wellness Policy: A Health Promoting Schools Assessment

**DOI:** 10.3390/ijerph21101372

**Published:** 2024-10-17

**Authors:** Matthew Chrisman, Anita Skarbek, Patricia Endsley, Nicholas Marchello

**Affiliations:** 1School of Nursing and Health Studies, University of Missouri-Kansas City, Kansas City, MO 64108, USA; skarbeka@umkc.edu; 2School of Education and Counseling, Cambridge College, Boston, MA 02129, USA; pendsley@wocsd.org; 3Department of Nutrition, Kinesiology, and Health, University of Central Missouri, Warrensburg, MO 64093, USA; marchello@ucmo.edu

**Keywords:** school wellness, teachers, principals, school nurses

## Abstract

The current study explored teacher and principal familiarity with school wellness polices in primary schools, including who serves on school wellness committees, and who should implement and enforce wellness policies in the school. An electronic survey guided by the Health Promoting Schools framework was administered from February to May 2020 to teachers and principals from one urban and one suburban school district in the Midwestern United States. There were 450 participants; response rates were 28% (urban), 33% (suburban), and 51% (school principals). Only 41.7% of the aggregate participant pool were familiar with their wellness policy. Participants were more familiar in the suburban compared to the urban district (χ^2^ = 68.2, *p*-value ≤ 0.001). Teachers/health teachers, nurses, and principals were most likely to be on wellness committees, and the most preferred wellness champions were teachers, nurses, and food service staff. Teachers and nurses are integral to school wellness and health education as part of multiple systems that can promote school health. The Health Promoting Schools framework is useful for guiding examinations to improve understanding of school wellness within school communities.

## 1. Background

It is recognized that school systems are integral in promoting optimal student health through wellness and education initiatives. Wellness includes (a) employing healthy habits, including good nutrition/physical activity behaviors, (b) conscientiousness, which requires transitioning from awareness to goal-directed behavior, and (c) intentionally aiming to achieve an optimum state of health [1]. The Healthy, Hunger-Free Kids Act [2] mandates that public schools participating in the National School Lunch or Breakfast Programs are to provide whole grains, fruits and vegetables, lean protein, and low-fat dairy options while limiting sugar and sodium. It further requires participating schools to adopt wellness policies that focus on nutrition education, wellness-promoting activities, designating faculty/staff responsible for policy oversight, student and community engagement, and systematic assessment from design to implementation [2].

In the United States, according to the Centers for Disease Control and Prevention (CDC) [3], “a local school wellness policy is a written document that guides a local educational agency (LEA) or school district’s efforts to create supportive school nutrition and physical activity environments”. Creation, application, and revision of local wellness policies should be a collaborative effort between school administrators, the local school board, schoolteachers, school health professionals, students, parents, and the community and should be reviewed, revised if needed, and publicly disseminated every three years [3]. Previous studies exploring the quality of local school wellness policies found inconsistencies in compliance and overall vagueness in policy language rather than providing specific details on how the policy meets the federal requirements [4]. The Health-Promoting Schools (HPS) framework encourages a holistic, whole-school approach to health and educational attainment in the school setting [5]. In a nationwide sample studying compliance of school wellness policies to legislation requirements, nutrition education was the most consistently addressed component, while regulation of on-campus food offerings was the weakest [4].

The CDC Healthy Schools program promotes the *Whole School Whole Community Whole Child* (WSCC) model; this model is an example of an HPS framework that many schools use to supplement wellness policies, as it includes nutrition, physical activity, family/community engagement, mental health, and environmental safety [6]. The advantages of the WSCC model are that it is student-centered and highlights the role of the community as well as evidence-based practices [6].

Primary and secondary schools play a critical role in physical and social facilitation of positive health behaviors, as children spend a significant amount of time at school [4]. Through wellness policies, schools can establish environments that promote the development and maintenance of healthy body mass indexes (BMIs) by offering nutritious meals, increasing opportunities for physical activity, and providing appropriate education on healthy eating and exercise [4]. Utilization of this scaffolding structure as a framework provides opportunities to systematically transform the physical and social environments of the *institution* rather than only promoting change in *individual* children. Lack of awareness and knowledge of existing local wellness policy, however, appears to be a barrier to successful implementation, support, and enforcement [7].

As teachers and principals play a significant role in school systems, their awareness, knowledge, and perspectives are key in promoting optimal school wellness [8].

## 2. Purpose

The purposes of this study were three-fold: (1) to determine teachers’ and principals’ familiarity with their school district’s wellness policy and wellness committee; (2) to determine their familiarity with their wellness committee’s duties and member roles; and (3) to determine who would best serve to oversee compliance with their school’s/district’s wellness policy as wellness policy enforcers.

## 3. Methods

### 3.1. Sampling

This cross-sectional, observational study utilized an electronic survey distributed to teachers and principals in one urban and one suburban school district in the Midwestern United States. The urban school district represents 10 schools comprising 530 staff members (including 409 teachers) overseeing approximately 5500 students. The suburban school district represents 29 schools comprising 1218 staff members (including 967 teachers) overseeing approximately 14,000 students [9]. All teachers and principals were eligible to participate. Both school districts were part of the Community Eligibility Provision, meaning all students enrolled were eligible for free school meals.

### 3.2. Instruments

Participants were asked six questions regarding their school wellness policy (Table 1). Utilizing a modified version of an electronic survey administered through Research Electronic Data Capture (REDCap), these six questions pertaining to wellness policy were a portion of a larger instrument consisting of 104 total questions. In addition to wellness policy awareness, the larger instrument also assessed demographics and the awareness, use, and knowledge of MyPlate nutritional guidelines in teaching. Estimated survey completion time was 15 min. The HPS framework provided initial guidance in survey question development [5]. The questions in the larger instrument address nutrition instruction in the classroom for teachers and the support principals provide to their schools and teachers regarding nutrition education. Nutrition professionals contributed to the development of the instrument, which was pretested among schoolteachers to ensure face and content validity.

### 3.3. Procedure

The instrument was modified to include the additional six questions related to local school wellness policies and committees, which were pretested among superintendents in neighboring school districts to ensure face and content validity. To assess test–retest reliability of the entire modified survey, 26 teachers in a separate sample completed the survey twice, with an average of 26 days between surveys. All variables had kappa values between 0.45 and 0.90, indicating moderate to substantial agreement.

All aspects of this study were reviewed and approved by an ethical and institutional review board (IRB) at the University of Missouri-Kansas City. The IRB approved the exempt status (IRB #2019185) of this study, and therefore informed consent was not collected but was implied by survey completion. Upon approval by the IRB, school district superintendents were contacted via email and asked to approve their staff to participate in the study. Once written approval was obtained, a member of the research team emailed the REDCap survey link to the school district’s teachers and principals. A single participation reminder was emailed to the same populations. Participants were asked to complete the survey on their own time to ensure anonymity and privacy, and data are only reported in the aggregate to ensure confidentiality. Data were collected from February to May 2020. Teachers and principals who completed the survey received a USD 20 electronic gift card.

### 3.4. Data Analysis

Descriptive data are presented as frequencies and percentages for categorical variables and means with standard deviations (SDs) for continuous variables. Analyses examining differences among groups, including demographic variables, urban vs. suburban, school level, and familiarity with school wellness policy (yes, no, or I don’t know), were conducted to compare participants’ characteristics using Pearson’s chi-square (χ^2^) test of independence for categorical variables. Statistical analyses using SPSS version 26.0 (IBM Corp., Armonk, NY, USA) were conducted on all available non-missing cases. Missing data were treated using pairwise deletion. Statistical tests were deemed significant at *p* < 0.05.

## 4. Results

The sample (n = 450) included 421 teachers, 20 principals, and 9 undisclosed participants. The response rate for the urban school district was 28.2%, while the suburban school district’s response rate was 33.0%. Reasons for non-response were not assessed due to resource limitations. Throughout both school districts, a total of 51.3% of principals (20 out of 39) participated. The age range of respondents was 22–68 years (mean age: 40.2). The sample was predominantly female (84.4%, n = 380). A majority of participants (67.3%, n = 303) were married or in a domestic partnership. Additional demographic characteristics are shown in Table 2. According to the state department of education, teachers average 12.6 years of teaching experience, and 59% have a Master’s degree or higher [10]. Our sample averaged 12.2 years of teaching experience, while 67.1% held a Master’s degree or higher. Table 3 shows how state and district location statistics compare to the two school district samples in this study. The average years spent teaching was comparable across the state, the districts, and our samples; however, our participants had a higher proportion of those with a Master’s degree or higher compared to their respective district and the state overall.

### 4.1. Familiarity

Table 4 shows participant familiarity with their district wellness policy and committee according to selected characteristics. In the overall sample, 41.7% were familiar with their district wellness policy. Participants were more familiar in the suburban district [n = 148 (53.8%)] compared to the urban district [n = 7 (9.9%); χ^2^ value: 68.2, *p*-value ≤ 0.001]. Those who stated they were married [n = 111 (71.6%) were more likely to be familiar than those who were not married [n = 42 (28.5%); χ^2^ value: 14.2, *p*-value = 0.001). Elementary school participants were more likely to be familiar with wellness policies than the middle or high school participants (44.7% elementary compared to 43.8% middle and 33.3% high school; χ^2^ value: 15.9, *p*-value = 0.003).

### 4.2. Wellness Committees

Participants who responded “yes” (n = 170) to being familiar with their district wellness policy were then asked an additional question regarding who serves on their wellness committees and how often the committee meets. When asked who serves on their local wellness committee, the most common responses were teachers [general; n = 83 (48.8%)]; nurses (n = 52; 30.6%); teachers [health-related; n = 36 (21.2%)]; principals (n = 35; 20.6%); and food service staff (n = 24; 14.1%). The least common responses were school board members (n = 6; 3.5%); members of the public (n = 5; 2.9%); county health officials (n = 2; 1.2%); and students (n = 2, 1.2%). Fifty-nine participants (34.7%) did not know who served on their wellness committee. Regarding how often wellness committees meet during the year, the most common responses were once per month (n = 26; 15.3%); once per quarter (n = 12; 7.1%); once per semester (n = 5; 2.9%); and once per year (n = 1; 0.6%). A majority (n = 110, 64.7%) did not know how often their wellness committees met during the year.

### 4.3. Wellness Champions

When all participants were asked to choose the top three individuals they thought should serve as a wellness champion or enforcer, the most frequent responses were nurses (n = 264; 58.7%); food service staff (n = 208; 46.2%); teachers [general; n = 168 (37.3%)]; teachers [health-related; n = 167 (37.1%)]; and local wellness committee coordinators (n = 126; 28.0%). Participants were then asked via an open-ended question to rank which individual was most important as a wellness committee enforcer or champion of their top three choices. Table 5 displays the individuals that participants perceived as the most important as a wellness champion or enforcer among their top three. School nurses (22.7%); health-related teacher, coaches, or some combination of those (15.1%); general teachers (11.6%); and the wellness coordinator (11.1%) were the most frequently preferred enforcers. Students, community members, and public health officials were preferred least at 1% each.

## 5. Discussion

This study assessed the familiarity of local school wellness policies and wellness committee existence among teachers and principals employed in urban and suburban school districts in a single midwestern state in the United States. Since the development and implementation of school-based wellness policies aligns with the Healthy People 2030 nutrition and healthy eating goal to improve health by promoting healthy eating behaviors and access to nutritious foods [11], this study is timely and may inform future research in this area, as well as help develop strategies to promote school wellness. Further, the findings support the HPS’s use as a guide to examine wellness in schools from a more holistic approach, involving not only teachers and principals, but additional staff such as school nurses and food service staff. Response rates for teachers in this study were comparable to previous studies [8]; however, there was greater participation in this study as compared to previous school district studies conducted by our team in this region [12].

### 5.1. Familiarity with District Wellness Policies

There was low familiarity with wellness policies and committees. This is concerning, as low familiarity impacts the implementation and enforcement of wellness policies [7]. Conversely, teacher awareness of wellness-related policies correlates with increased policy implementation [13].

For this study, familiarity with wellness policies and committees was higher in the suburban school district as compared to the urban district, suggesting that work may be needed to improve awareness in urban districts. The reasons for urban–suburban differences are unclear, and much of the existing literature compares urban and rural districts [14,15]. We postulate that the differences are due to suburban schools having more resources available in general. In fact, the suburban school district that participated in this study had a dedicated school district wellness coordinator, who could enhance familiarity via regular communication (e.g., emails, newsletters, etc.) and district events (step contests).

Although there is a paucity of research regarding wellness policy familiarity, an analysis of the 2016 CDC *School Health Policies and Practices Survey* data illustrated that urban districts had more general health policies and dissemination, but less policy application, than suburban school districts [16]. Recommendations to improve the awareness and implementation of wellness policies have been in existence for over a decade [13]; these include involving teachers in policy development, providing continual reminders of wellness policy existence, and providing ongoing health education development that promotes policy awareness, implementation, and assessment. Additional research is needed to examine current evidence-based strategies to improve awareness of wellness policies in schools, as well as to examine this topic in rural schools.

### 5.2. School Wellness Committees

Concurrent with low familiarity with wellness policies, >50% of participants were unable to identify who served on their district/school wellness committees; however, school nurses and teachers were the most highly recognized committee members. Since there is limited evidence to support the perception that school nurses are the most commonly perceived policy enforcers, the National Association of School Nurses (NASN) contends that school nurses, as system-level leaders, are uniquely qualified to take a lead role in school wellness policy development and implementation due to their specialized education and training [17,18]. The American Nurses Association (ANA) asserts that a primary role for school nurses includes nutrition program support to promote student healthy eating habits [19].

The finding that most wellness committees met more than once per year is encouraging given evidence that students of schools with wellness committees that meet at least once per year have lower BMIs, consume fewer sugar-sweetened beverages, and consume breakfast more frequently than those in schools with wellness committees that do not meet or in schools that do not have a wellness committee [20].

### 5.3. Wellness Policy Champions

There is a gap in the literature concerning wellness policy enforcement and who should assist in this role. However, it has been demonstrated that school districts need to designate wellness coordinators who are responsible for implementing, enforcing, evaluating, and educating about district wellness policies [7]. In accordance with Title I and Title IV regulations, the CDC [21] stipulates that school districts are required to permit a wide variety of individuals, including parents, students, representatives of the school food authority, physical education teachers, school health professionals, school board members, school administrators, and the general public, to participate in the development, implementation, and update of local school wellness policy.

Teachers and nurses were perceived in this study as the most important potential wellness policy enforcers. Although schoolteachers and nurses often do not have the authority to act as wellness policy enforcers or champions, they are well positioned to serve in this role [18]. Programs such as the national nutritional guidelines (MyPlate), when enmeshed in school culture through policy and environmental levels of influence, offer opportunities for schoolteachers and nurses to initiate conversations and be role models on a personal level of influence [22]. The WSCC is a best practice model for integrating health and wellness into education programs [23]; it expands traditional health education to other classroom and community settings and emphasizes the nutrition environment and health education as two core components. Applying a WSCC approach emphasizes the need for coordinated policies and practices [24], highlighting how wellness policy enforcement is critical to supporting learning and health. The WSCC model is student-centered; given that students were among the least preferred wellness policy enforcers here, there is potential for promoting greater student involvement in school wellness.

Further, in the Framework for School Nursing Practice™, NASN asserts that school nurses are positioned to lead the development of programs, policies, and procedures for improving student health [17]. Exercising leadership as part of the school environment of the HPS framework enables school nurses and health-related teachers to comprehend and address the integral relationship between the learning environment, educational stakeholders, and institutional policies and procedures [25]. School nurses have historically advocated for nutrition education and are theoretically well positioned through their expertise and visibility in the school setting to collaborate with health educators and promote wellness policy initiatives.

For example, one New Jersey school nurse leads an award-winning school vegetable garden project and has sustained the project’s success [26]. Despite these recommendations and opinions, the workload of perceived wellness champions must also be considered. Currently, school nursing has no validated workload determination tool. One school nurse may cover multiple buildings and thousands of students, thus preventing participation on wellness committees [18]. Regional differences may also impact champion involvement. A recent school nurse workforce report indicates that 88.2% of schools employ full-time school nurses compared to 59.9% of schools in the Midwest [27]. Superintendent turnover rate has also increased. The stress of managing day-to-day operations in addition to student learning needs and employee engagement (post-COVID-19) may take priority over wellness and other committees [28]. Interestingly, in this study, principals were listed as preferred wellness policy enforcers by only 3.3% of participants, suggesting that they may be viewed by other participants as playing more of a supporting role in school wellness.

A recent study examined national school district wellness policies that were available online and concluded that, overall, wellness policies were “mediocre”, “not comprehensive”, and did not address all current federal regulations [29]. Involvement of champions or staff committed to school health and wellness culture may be beneficial to the development of a system that promotes comprehensive wellness policy development and compliance. As an example of what this might look like in practice, Chicago Public Schools created a Wellness Champion Leadership Council that meets monthly to discuss new initiatives for fundraisers, celebrations, and wellness culture district-wide [30]. Further, Boston Public Schools also offer a Wellness Champion Program. Interested staff attend a formal orientation session and are offered professional development, resources, and a stipend. Champions must remain active in their school wellness committees and complete evaluations and reports.

### 5.4. Limitations and Strengths

The cross-sectional nature of the data, which inhibits causal inferences in terms of understanding familiarity with school wellness, was a limitation of this study. Also, the participants in our sample reported higher education than their respective district and the state on average, and thus may not reflect teachers in the state overall. This study did not examine how many years teachers and principals had worked in their particular school, which could affect their familiarity with wellness policies and committees. However, a strength of the study was the large sample size, allowing group comparisons. This contributes to an initial assessment of school nutrition education and wellness policy enforcement in multiple school districts, particularly in metropolitan areas that include both urban and suburban school districts.

## 6. Conclusions

Familiarity with school wellness policy is low among schoolteachers, suggesting a lack of effective processes to promote awareness and enforcement of the policy. Schools with wellness policies in place have enhanced wellness environments [31]. Increasing teachers’ familiarity with their wellness policy and committee is one strategy to increase wellness activities, and thus improve the wellness environment in schools. The purpose of school-based wellness policies is to promote the overall health and well-being of the population in the school setting [32]. An HPS approach could assist in identifying factors causing a lack of awareness and in further exploring improvement and sustainment strategies. As health-related teachers and school nurses are influential in health education and wellness, they may be well suited to promote school wellness policy and serve as content resources for other staff. Additionally, integrating wellness concepts into other core classes supports the best practice WSCC and HPS models. All of these recommendations must take into account wellness champions’ current workload.

Our findings demonstrate that teachers and school nurses play a critical role in school wellness and health education, which is comparable to other findings [31]. However, over half of the participating teachers were unaware of their district’s wellness policy, suggesting a dysfunction in the process of policy awareness, promotion, and enforcement. Thus, there is also a need to further investigate policy familiarity in schools, including the classroom and community environments, per the HPS framework, with particular attention to urban schools, and to examine the potential implications for the workload of individual wellness champions and policy enforcers. Measurement of wellness policy promotion and enforcement is achievable through the mandated triennial compliance assessment report [21], which school nurses and health teachers could contribute to and support. This report should be disseminated to the public to establish collaboration between the school district, its schools, parents, and the community.

## Figures and Tables

**Table 1 ijerph-21-01372-t001:** Wellness policy and committee questions and response options.

Question	Response Options
Does your school have a wellness committee?	Yes, No, Don’t know
Who serves on the school wellness committee?	Select all that apply *
What does your school wellness committee do?	Select all that apply ^
How frequently does the local school wellness committee meet?	Once per week, once per month, once per quarter, once per semester, once per year, don’t know
Who do you think would be the best individual(s) to serve as a wellness champion or enforcer of the school wellness policy?	Select up to 3 *
Of those top three individuals, who do you think would be the best individual to serve as a wellness champion or enforcer of the school wellness policy?	Open-ended response

* Options included: teachers, principal, superintendent, coach, physical education teacher, health education teacher, food service staff, food service director, school nurse, parents, and community members. ^ Options included: assists with policy development or revision to support a healthy school environment; promotes parent, community, and professional involvement in developing a healthier school environment; taps into funding and leverages resources for students and staff; plans and implements wellness programs for students and staff; evaluates wellness program and policy efforts; provides feedback to the district regularly regarding progress on the implementation of the local wellness policy; regularly reports on content and implementation to the public (including parents, students, and community members); periodically measures school compliance with the local wellness policy and progress updates made available to the public; don’t know.

**Table 2 ijerph-21-01372-t002:** Self-reported demographic characteristics of participants.

Variable	n (%)
**Age**, mean (SD)	40.24 (11.3)
**Age Category** n. (%)	
20–29	96 (21.3)
30–39	135 (30.0)
40–49	111 (24.7)
>50	107 (23.8)
**Sex** n. (%)	
Female	380 (84.4)
Male	64 (14.2)
**Ethnicity** n. (%)	
White	376 (83.6)
Black	45 (10.0)
Hispanic	11 (2.4)
Other	13 (2.9)
**Marital Status** n. (%)	
Married	274 (60.9)
Not married	175 (39.0)
**Health Status** n. (%)	
Fair	66 (14.7)
Good	216 (48.0)
Very good	133 (29.6)
Excellent	32 (7.1)
**BMI** n. (%)	
Normal	116 (25.8)
Overweight	125 (27.8)
Obesity	198 (44.0)
**School District** n. (%)	
Suburban	329 (73.1)
Urban	118 (26.2)
**School Level** n. (%)	
Elementary	234 (52.0)
Middle	100 (22.2)
High	116 (25.8)
**Education**	
Bachelor’s degree	134 (29.8)
Master’s degree	296 (65.8)
Doctoral degree	9 (2.0)
**Years of Teaching, mean (SD)**	12.2 (9.1)
**Years of Teaching** n (%)	
<10	196 (46.9)
10–19	127 (30.4)
>20	95 (22.7)

**Table 3 ijerph-21-01372-t003:** Comparison of participating urban and suburban school districts with teachers statewide.

Category	Statewide Total	Study Total Sample	Overall Urban District	Study Urban Sample	Overall Suburban District	Study Suburban Sample
Average years of teaching experience	12.6	12.2	10.9	11.9	11.3	12.9
Percentage of teachers with Master’s degree or higher	59%	67.1%	50.9%	66.4%	60.1%	67.8%

**Table 4 ijerph-21-01372-t004:** A comparison of participant demographic characteristics based on their familiarity of wellness policies and committee, using Pearson’s chi-square test (*t*-test was used for age as continuous variable).

Characteristics	Familiarity of Wellness Policies and Committee			*p*-Value
	Yes (n = 155)	No (n = 90)	I Don’t Know (n = 127)	
**Age**, mean (SD)	40.7 (10.9)	40.7 (12.2)	39.5 (10.6)	0.57
**Age Category** n. (%)				0.17
20–29	30 (19.4)	25 (27.8)	22 (17.3)	
30–39	47 (30.3)	19 (21.1)	48 (37.8)	
40–49	43 (27.7)	23 (25.6)	28 (22)	
>50	35 (22.6)	23 (25.6)	29 (22)	
**Sex** n. (%)				0.11
Female	130 (83.9)	82 (91.1)	102 (80.3)	
Male	22 (14.2)	8 (8.9)	24 (18.9)	
**Ethnicity** n. (%)				0.17
White	137 (88.4)	74 (82.2)	107 (83.4)	
Black	9 (5.8)	13 (14.4)	12 (9.4)	
Other	9 (5.8)	3 (3.3)	8 (6.4)	
**Marital Status** n. (%)				0.001
Married	111 (71.6)	44 (48.9)	74 (58.3)	
Not married	42 (28.5)	46 (51.1)	49 (41.7)	
**Health Status** n. (%)				0.22
Fair	19 (12.3)	12 (13.3)	23 (18.1)	
Good	67 (43.2)	41 (45.6)	66 (52)	
Very good	53 (34.2)	27 (30)	31 (24.4)	
Excellent	16 (10.3)	9 (10)	6 (4.7)	
**BMI** n. (%)				0.79
Normal	43 (27.7)	20 (22.2)	27 (21.3)	
Overweight	45 (29.0)	27 (30.0)	36 (28.3)	
Obesity	63 (40.6)	37 (41.1)	59 (46.5)	
**School District** n. (%)				<0.001
Suburban	148 (95.5)	47 (52.2)	80 (63)	
Urban	7 (4.5)	41 (45.6)	47 (37)	
**School Level** n. (%)				0.003
Elementary	85 (54.8)	56 (62.2)	49 (38.6)	
Middle	39 (25.2)	17 (18.9)	33 (26)	
High	31 (20)	17 (18.9)	45 (35.4)	
**Years of Teaching** n. (%)				0.18
<10	56 (36.1)	42 (46.7)	61 (48)	
10–19	51 (32.9)	21 (23.3)	39 (30.7)	
>20	33 (21.3)	19 (21.1)	19 (15)	

**Table 5 ijerph-21-01372-t005:** Participants’ top 5 most frequently perceived wellness champions.

Role/Position	n (%)
Nurse	102 (22.7%)
Health Teacher and/or Coach	68 (15.1%)
Teacher	52 (11.6%)
Wellness Coordinator	50 (11.1%)
Food Service Staff	39 (8.7%)

## Data Availability

Data is available by contacting the corresponding author.

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
