# Peer review of "Teachers’ and Principals’ Familiarity with School Wellness Policy: A Health Promoting Schools Assessment"

_ijerph, 2024, doi:10.3390/ijerph21101372_

Round 1
Reviewer 1 Report
Comments and Suggestions for Authors
Matthew Chrisman et al. submitted to IJERPH an article, dealing with an investigation performed in primary schools, exploring teacher and principal familiarity regarding school wellness polices.
This study is very dated, it refers to data collected from February to May 2020, I would say quite obsolete and no longer adherent to current reality. It certainly needs to be implemented, taking into account a total new investigation.
Consequently, the discussions need to be rewritten taking into account the new findings that have emerged.
The references are not cited according to the Journal rules.
Comments on the Quality of English LanguageMinor editing of English language required.
Author Response
We have replied to all reviewer comments in the attached table. Thank you for the critical feedback.

Reviewer 2 Report
Comments and Suggestions for Authors
First of all, I would like to congratulate the authors for the choice of the study topic. Secondly, I would like to ask you to carefully check all the bibliographical references for compliance with the standard not only in the development of the article, but also in the final list. For example: the first reference Skarbek et al., 2023 does not exist in the final list; the reference identified in line 21 is labelled differently in the final list with the number 1. Thirteen references were identified in the final list without a match (2,3,4,5,7,8,16,27,31,32,35,36,38). The authors should review all these aspects and remove the final numbering, as the references comply with APA.
According to the study chapters:
1 - The Background as no issues to improve.
2 - The Methods chapter can be improved on the 2.2, 2.3 and 2.4 sections:
2.2 - Please correct on the line 92 the "seven questions" for six questions, so that in line with the presented. Must also respect the formatting of the observations (*) included in table 1 when changing pages. Lines 112-118 are confusing for the reader who may assume that they are text and not the observations of table 1.
2.3 - Must be improved and clarified the request and opinion of an ethics committee. Being a study involving humans, these aspects must be ensured and highlighted in the methodology chapter. All procedures for informed consent must be specified, as well as ensuring the anonymity of participants.
2.4 - The pointed choice of the statistical test is not enough and needs clarification. What specific comparisons were made? Was a null hypothesis stipulated? What type of comparison was made? Adherence test? homogeneity? independence? The choice of the chi-square test (Pearson) and the way in which it was used need to be clarified.
3 - On the results should be clear that your sample was not representative of the population studied and this a study limitation. According to your population (urban - 530 persons and suburban - 1218), 450 participants correspond to 25,74% of your population. As the study with the highest number of participants, this aspect should be emphasized and the conclusions restricted to the sample studied.
The data in table 3 also needs clarification. The comparison is not made with data in the same valence. Some are in percentages and others are not. The table should complement with what is written, and not the other way round.
The evidence from the chi-square test is not identified in table 4. The authors should present the analysis carried out and the comparison of variables studied.
In Table 5, the total n (%) does not add up to 450. Complete with all the data or only fulfil the top 3 as described in the text.
4 - On the limitations and strengths, the authors must present the limitations of the study more precisely.
5 - On the conclusions chapter the authors should highlight the main findings of the study and emphasize their importance. It's too general and comprehensive.
Author Response

(The authors gave the same response as above.)

Reviewer 3 Report
Comments and Suggestions for Authors
I appreciate the opportunity to review this document. I believe the study is interesting and carefully conducted, making it suitable for publication in the journal, provided that some modifications, which I specify below, are made to improve its clarity and format.
The BACKGROUND of the article presents a solid focus on the importance of school wellness policies and their impact on student health. This introduction provides a strong foundation for the study and establishes a relevant context. However, it's necessary to modify the format of the figure because it is not legible.
In the PARTICIPANTS section, more information about the participants is needed. The information appearing at the beginning of the results section (those corresponding to Tables 2 and 3) is a description of the participants' sociodemographic characteristics, not the actual study results, and therefore, should be included in the Participants section.
In the INSTRUMENT section, valuable information is provided about the survey content, including specific questions regarding school wellness policy and the use of a modified electronic survey. However, there is a mix of details about the development and validation of the instrument that might confuse the reader. The description of the questionnaire questions and their response options is provided in detail in this section. Although this is useful, the information on how these questions were administered and how the responses were collected should be clearly linked to the study’s administration procedure. Therefore, the Instrument section should focus on describing the content of the questionnaire and the questions used, while the PROCEDURE section should detail the entire process of administration, data collection, and validation. Moving the details about the instrument's validation, pre-testing, and reliability assessment to the Procedure section will allow the reader to clearly understand how the instrument was developed and tested before its actual administration.
The RESULTS section provides a clear and detailed presentation of the data, breaking down participant responses by categories such as familiarity with the wellness policy, composition of wellness committees, and perceptions of wellness champions. One aspect that needs modification is that although a comparison is made between suburban and urban districts, a deeper analysis of why significant differences exist is not provided, which could be relevant for the discussion.
The DISCUSSION addresses the importance of familiarity with wellness policies and committees, linking these findings to existing literature. Practical implications are discussed, and strategies to improve awareness and implementation of policies are suggested, adding value to the research.
The limitations of the study are acknowledged, such as the cross-sectional nature of the data and the lack of information on the participants' work experience. This acknowledgment of limitations is essential for a balanced discussion. Despite this, and following what was mentioned in the results section, the discussion should include a deeper interpretation of the data, particularly concerning the differences between urban and suburban districts. It would be useful to explore possible reasons for these differences and how they might influence policy implementation. Although the need for further research is mentioned, the discussion should be more specific about which aspects of future research are necessary and how the identified gaps could be addressed.
The CONCLUSIONS highlight the importance of familiarity with wellness policies and the need to improve education and awareness among school staff. The alignment with theoretical models and the mention of practical strategies for implementation are appropriate and useful. Additionally, a clear direction for future research is proposed, and the implications for practice are emphasized. The assertion made about the need to further investigate familiarity with the policy and examine the implications for the workload of wellness champions should be more specific and detailed. Finally, the conclusions should emphasize the connection of the study’s findings with existing literature, especially in terms of how the results align or contrast with previous research and what new perspectives they bring to the field.
Author Response

(The authors gave the same response as above.)

Round 2
Reviewer 1 Report
Comments and Suggestions for Authors
Despite clarifications provided during the review round, this remains an outdated investigation paper. The authors improperly attempt to justify this largely non-current timing.
Authors must indicate in the abstract the year in which the data were collected.
References must be indicated according to the Journal rules.
Comments on the Quality of English LanguageMinor editing of English language required.
Author Response
We have addressed the reviewer comments in the attached document. Thank you for your critical feedback.

Reviewer 3 Report
Comments and Suggestions for Authors
The authors have taken into account all the suggestions made, so I consider that the article is ready to be published.
Author Response
We have addressed the reviewer comments in the revised manuscript. Thank you for your critical feedback.